# Stimulating the Hematopoietic Effect of Simulated Digestive Product of Fucoidan from *Sargassum fusiforme* on Cyclophosphamide-Induced Hematopoietic Damage in Mice and Its Protective Mechanisms Based on Serum Lipidomics

**DOI:** 10.3390/md20030201

**Published:** 2022-03-09

**Authors:** Wei-Ping Ma, Shi-Ning Yin, Jia-Peng Chen, Xi-Cheng Geng, Ming-Fei Liu, Hai-Hua Li, Ming Liu, Hong-Bing Liu

**Affiliations:** 1Key Laboratory of Marine Drugs, School of Medicine and Pharmacy, Ocean University of China, Qingdao 266003, China; maweiping1990@163.com (W.-P.M.); 21180831069@stu.ouc.edu.cn (J.-P.C.); 21200811111@stu.ouc.edu.cn (X.-C.G.); 21200831185@stu.ouc.edu (M.-F.L.); shaixuan@ouc.edu.cn (H.-H.L.); 2Qingdao Institute for Food and Drug Control, Qingdao 266000, China; yinshining@126.com; 3NMPA Key Laboratory for Quality Research and Evaluation of Marine Traditional Chinese Medicine, Qingdao 266000, China; 4Laboratory for Marine Drugs and Bioproducts, Pilot National Laboratory for Marine Science and Technology, Qingdao 266237, China

**Keywords:** *Sargassum fusiforme* fucoidan, hematopoietic damage, lipidomics

## Abstract

Hematopoietic damage is a serious side effect of cytotoxic drugs, and agents promoting hematopoiesis are quite important for decreasing the death rate in cancer patients. In our previous work, we prepared the simulated digestive product of fucoidan from *Sargassum fusiforme*, DSFF, and found that DSFF could activate macrophages. However, more investigations are needed to further evaluate whether DSFF could promote hematopoiesis in the chemotherapy process. In this study, the protective effect of DSFF (1.8–7.2 mg/kg, i.p.) on cyclophosphamide-induced hematopoietic damage in mice and the underlying mechanisms were investigated. Our results show that DSFF could restore the numbers of white blood cells, neutrophils, and platelets in the peripheral blood, and could also retard bone marrow cell decrease in mice with cyclophosphamide-induced hematopoietic damage. UPLC/Q-Extraction Orbitrap/MS/MS-based lipidomics results reveal 16 potential lipid biomarkers in a serum that responded to hematopoietic damage in mice. Among them, PC (20:1/14:0) and SM (18:0/22:0) were the key lipid molecules through which DSFF exerted protective actions. In a validation experiment, DSFF (6.25–100 μg/mL) could also promote K562 cell proliferation and differentiation in vitro. The current findings indicated that DSFF could affect the blood cells and bone marrow cells in vivo and thus showed good potential and application value in alleviating the hematopoietic damage caused by cyclophosphamide.

## 1. Introduction

Cytotoxic drugs are the first-line drugs used in the clinical treatment of malignant tumors [1], but they cause some serious side effects, such as hematopoietic damage [2], bone marrow suppression [3], and immunosuppression [4], which significantly increase the risk of death in cancer patients. At present, G-CSF and GM-CSF are clinically used white blood cell-stimulating agents [5]. However, when used for a long time, these colony stimulating factors also cause splenomegaly [6] and increase the risk of splenic rupture [7]. Therefore, adjuvant drugs with safe and effective hematopoietic damage protective functions are urgently needed.

The polysaccharides found in traditional Chinese medicines have been found to protect against the hematopoietic damage caused by cytotoxic drugs and even enhance the immune state of the body [8]. Fucoidans are a class of fucose-rich sulphated polysaccharides. The classical bioactivities of fucoidan include anti-oxidant, anti-tumor, anti-coagulant, and immunoregulation properties [9]. Most importantly, it has been reported that fucoidan stimulates hematopoiesis. For example, sea cucumber fucoidan (*Holothuria Polii*) could restore the number of white blood cells reduced by cyclophosphamide and significantly promote the recovery of neutrophils [10]; brown algae (*Chordaria flagelliformis*) fucoidan was also shown to promote the recovery of the white blood cell count in mice with cyclophosphamide-induced leukopenia [11]. In our previous studies, we prepared a fucoidan (SFF) and its simulated digestive product, fucoidan (DSFF), from *Sargassum fusiforme* and found that DSFF had an immunomodulatory effect and exhibited stronger macrophage activation and phenotype polarization effects in vitro [12]. It was considered that immune cells, especially macrophages, could promote the development [13,14] and reconstruction [15] of hematopoietic stem cells. In addition, changes in various lipid molecule levels have been observed in hematopoietic abnormalities and might be considered markers for disease monitoring and treatment [16]; therefore, they might affect peripheral blood hematopoietic stem cell levels [17]. Additionally, *Sargassum fusiforme* fucoidan was also reported to regulate lipid metabolism levels [18]. We hypothesized that the macrophage activation, the phenotype polarization effect, and the lipid regulation of DSFF have the potential to alleviate chemical drug-induced hematopoietic damage. Therefore, it is interesting and worthwhile to investigate the hematopoietic damage protection of DSFF in vivo and its protective mechanisms.

In this study, to systematically study the hematopoietic damage protection of DSFF and the overall related mechanism, the potential protective effect of DSFF on mice with hematopoietic damage was evaluated using the cyclophosphamide-induced hematopoietic damage mouse model. The effects of DSFF on the counts of white blood cells, neutrophils, and platelets, and the DNA content of bone marrow were evaluated. Moreover, we tried to clarify the potential mechanism affected by DSFF through screening of endogenous metabolite biomarkers based on the serum lipidomics method.

## 2. Results

### 2.1. Protective Effect of DSFF on Body Weight and Immune Organ Index in Hematopoietic Damage Mice

A single high-dose injection of cyclophosphamide has been reported to establish a cyclophosphamide-induced hematopoietic injury model, which can be used to evaluate the protective effect of drugs or components on hematopoietic damage [19,20]. In our present studies, we also established the cyclophosphamide-induced hematopoietic damage model by the intraperitoneal injection of 250 mg/kg cyclophosphamide in mice. As shown in Figure 1A, the body weight of the control group mice gradually increased throughout the experiment (Appendix A). However, compared with the control, a significant inhibition of body weight gain was observed in the model group mice induced by cyclophosphamide throughout the experiment (*p* < 0.01), which indicated the successful establishment of the hematopoietic damage model. Compared with the model, the inhibition of body weight gain in the 3.6 mg/kg DSFF group mice showed an alleviated trend from the 5th day onwards. Such as the 3.6 mg/kg DSFF group, compared with the model, the body weight gain inhibition in the rhG-CSF group mice also showed a trend of alleviation from the 5th day onwards. The cytotoxic drug-induced inhibition of body weight gain is usually considered to be an indicator of side effects [21]. These results indicate that DSFF might have a potential alleviation effect on the inhibition of body weight gain induced by cyclophosphamide.

Next, the effects of DSFF on splenic and thymic indexes were investigated. As shown in Figure 1B, compared with the control, the obviously increased splenic index induced by cyclophosphamide was observed in the model group (*p* < 0.01), which might be associated with compensatory organ enlargement caused by cyclophosphamide-induced reduced peripheral blood cytopenia [22]. Compared with the model group, the splenic index of the 1.8 and 3.6 mg/kg DSFF groups further increased significantly (*p* < 0.01). As a positive group, the splenic index of the rhG-CSF group was also markedly increased (*p* < 0.05). These results suggest that DSFF might have a strong stimulating effect on the enlarged spleen induced by cyclophosphamide.

For thymic index, compared with the control, the obviously decreased thymic index induced by cyclophosphamide was observed in the model group (Figure 1C, *p* < 0.01), which might be caused by the immunosuppression of cyclophosphamide. Compared with the model group, the decreased thymic index of the 1.8 mg/kg DSFF group was obviously enhanced (Figure 1C, *p* < 0.05). However, compared with the model group, the decreased thymic index of the rhG-CSF group had no change (Figure 1C, *p* > 0.05). These results suggest that DSFF might have a strong ameliorating effect on the thymus suppression induced by cyclophosphamide.

### 2.2. Effect of DSFF on Blood Routine in Cyclophosphamide-Induced Hematopoietic Damage Mice

#### 2.2.1. Restorative Effect of DSFF on White Blood Cell Count

Hematopoietic damage, including reduced white blood cells, neutrophils, and platelets, is one of the main side effects of cyclophosphamide [22]. As shown in Table 1, on the first day after administration, compared with the control group, the white blood cell count (WBC) of the model group mice was significantly decreased (*p* < 0.01), which further indicated the successful establishment of the cyclophosphamide-induced hematopoietic damage model. On the third day after administration, compared with the control group, the WBC of the model group mice was further decreased (*p* < 0.01). Compared with the model group, the WBCs of the DSFF group and rhG-CSF group mice had no difference. On the fifth day after administration, compared with the control group, the WBC of the model mice had still not recovered (*p* < 0.01). Compared with the model group, the WBCs of the 1.8, 3.6, and 7.2 mg/kg DSFF group mice significantly increased (*p* < 0.01). Compared with the model group, the WBC in the rhG-CSF group mice also significantly and rapidly increased (*p* < 0.01). On the seventh day after administration, the WBC in the model group mice showed no statistical difference from that of the control group (*p* > 0.05). However, compared with the model, the WBC in the 1.8 mg/kg DSFF group of mice further significantly increased (*p* < 0.01). Akin to the 1.8 mg/kg DSFF group, the WBC in the rhG-CSF group mice were also higher than that in the model group (*p* < 0.01). The above results suggest that DSFF, especially at 1.8 mg/kg of DSFF, could rapidly promote the decreased WBC of mice with cyclophosphamide-induced hematopoietic damage to return to a normal level.

#### 2.2.2. Restorative Effect of DSFF on Neutrophil Count

Hematopoietic damage caused by chemotherapeutic agents such as cyclophosphamide is mainly reflected in a decrease in neutrophils and lymphocytes [23]. Therefore, the effect of DSFF on the neutrophil count was investigated. As shown in Table 1, on the first day after administration, compared with the control group, the neutrophil count of the model group mice was significantly decreased (*p* < 0.01), which also indicated the successful establishment of the cyclophosphamide-induced hematopoietic damage model. On the third day after administration, compared with the control group, the neutrophil count of the model group mice was further decreased (*p* < 0.01). Compared with the model group, the neutrophil counts of the DSFF group and rhG-CSF group mice showed no difference. On the fifth day after administration, compared with the control group, the neutrophil count of the model mice was still not recovered (*p* < 0.01). Compared with the model group, the neutrophil counts of the 1.8, 3.6, and 7.2 mg/kg DSFF group mice were significantly increased (*p* < 0.01). Compared with the model group, the neutrophil count in the rhG-CSF group mice was also significantly and rapidly increased (*p* < 0.01). On the seventh day after administration, the neutrophil count in the model group mice showed no statistical difference from that of the control group (*p* > 0.05). However, compared with the model, the neutrophil count in the 1.8 mg/kg DSFF group mice further increased significantly (*p* < 0.05). Akin to the 1.8 mg/kg DSFF group, the neutrophil count in the rhG-CSF group mice was also higher than that in the model group (*p* < 0.05). The above results suggest that DSFF, especially at 1.8 mg/kg DSFF, could encourage the neutrophil count of mice with cyclophosphamide-induced hematopoietic damage to return to a normal level.

#### 2.2.3. Restorative Effect of DSFF on Platelet Count

Thrombocytopenia induced by chemotherapy drugs is the main cause of side effects such as gastrointestinal and visceral bleeding in cancer patients [24]. As shown in Table 1, on the third and fifth days after administration, compared with the control group, the platelet count of the model group mice was significantly decreased (*p* < 0.01). Compared with the model group, the platelet counts of the DSFF group and rhG-CSF group mice showed no difference. On the seventh day after administration, the platelet count in the model group mice was slightly lower than that of the control group (*p* > 0.05, Table 1). Compared with the model group, the platelet counts in the 1.8 and 7.2 mg/kg DSFF group mice were obviously increased (*p* < 0.05). Compared with the model group, the platelet count of the rhG-CSF group mice showed no difference (*p* > 0.05). The above results suggest that DSFF, especially at 1.8 mg/kg of DSFF, had the potential to promote the recovery of decreased platelet count in mice with cyclophosphamide-induced hematopoietic damage.

### 2.3. Protective Effect of DSFF on Bone Marrow DNA Content

Bone marrow suppression is also a major side effect of chemotherapy drugs [25]. As shown in Figure 2, compared with the control group, the bone marrow DNA content of the model group mice was reduced significantly (*p* < 0.01). Compared with the model group, the bone marrow DNA content of the rhG-CSF group mice was significantly increased (*p* < 0.01). As with the effect of rhG-CSF, the bone marrow DNA content in the 7.2 mg/kg of DSFF group mice was also significantly increased (*p* < 0.01), which indicated the ameliorating effect of DSFF on myelosuppression induced by cyclophosphamide.

### 2.4. The Hematopoietic Damage Protection Mechanism of DSFF Based on Serum Lipidomics

#### 2.4.1. Screening of Potential Lipid Biomarkers for Hematopoietic Damage in Mouse Serum

Due to the diverse structure of fucoidans, it is difficult to investigate the action mechanisms of DSFF in vivo. In the present study, we employed lipidomics methods to investigate the changes in lipid composition and concentration in the cyclophosphamide-induced hematopoietic model mice treated with DSFF and tried to illustrate the mechanisms underlying the protective effect of DSFF in cyclophosphamide-induced hematopoietic damage mice.

Firstly, the metabolite data obtained by the UPLC-MS/MS in positive and negative ion mode were imported into SIMCA for principal component analysis (PCA). The results showed that after an injection with 250 mg/kg of cyclophosphamide, the metabolites in the model group changed abnormally and were separated completely from the control group (Figure 3A,B). Moreover, the metabolites in the positive-drug group (rhG-CSF group) clustered away from the model group in PCA (Figure 3C,D), and almost overlapped with those of the control group (Figure 3C). The results indicate that the model of cyclophosphamide-induced hematopoietic damage in mice was replicated successfully.

OPLS-DA was performed to observe the metabolite clustering between the control group and the model group (Figure 4). Potential biomarkers were screened and identified by the comprehensive utilization of one-way analysis of variance, the Progenesis QI software (VIP > 1, *p* < 0.05), and some metabolic databases (Lipid Maps, HMDB, KEGG, and Massbank). As a result, 16 potential lipid biomarkers in serum that responded to hematopoietic damage in mice were found (Table 2), which were related to the pathways of glycerophospholipid metabolism, linoleic acid metabolism, alpha-linolenic acid metabolism, and arachidonic acid metabolism (Figure 5).

#### 2.4.2. Identification of Potential Lipid Biomarkers for DSFF Protective Effects

DSFF exhibited hematopoietic protection in the animal experiments, which was supported by the data from serum lipidomics. The PCA and OPLS-DA of metabolites in the control group, the model group, and the 3.6 mg/kg DSFF group showed the same results, as the points of each group gathered well and were distributed in different quadrants. Compared with the model group, the DSFF group showed an obvious trend of callback to the control group (Figure 6).

After treatment with rhG-CSF, the contents of three biomarkers, LysoPC (P-18:0), PC (20:3/18:0), and SM (d18:0/22:0), tended to be normal (Table 2). For DSFF, six potential lipid biomarkers were found. Two of them ((PC (20:1/14:0), SM (d18:0/22:0)) were therapeutic biomarkers, while the other four ((2-palmitoyl-sn-glycero-3-phosphocholine, PC (20:1/14:0), PC (20:5/16:0), and DG (18:0/20:4)) were potentially negative biomarkers (Table 2). These six biomarkers of DSFF involved all four pathways in the hematopoietic damaged mice model, while the underlying mechanism still needs to be studied.

### 2.5. Promoting Effect of DSFF on Proliferation and Differentiation of Myeloid K562 Cells

Next, to confirm the differential metabolic pathways predicted above, the effects of DSFF on the proliferation and differentiation of bone marrow cells were studied in vitro [26,27]. K562 cells are myeloid lymphoblasts with the ability to differentiate into erythrocytes, megakaryocytes, and monocytes; thus, they are usually used to study the hematopoietic recovery effect of drugs [28,29]. The results showed that DSFF could moderately promote the proliferation of the K562 cells (Figure 7A), suggesting that DSFF might promote the recovery of white blood cells by promoting the proliferation of bone marrow cells.

Then, the effects of DSFF on the differentiation of K562 cells were investigated. After the catalytic oxidation of benzidine by red blood cell hemoglobin [29], the produced emerald blue benzidine blue indirectly reflects the number of red blood cells [28]. Our results show that, after treatment with different concentrations (6.3–100 μg/mL) of DSFF, the percentage of benzidine-staining positive K562 cells significantly increased in a concentration-dependent manner, demonstrating the role of DSFF in inducing the differentiation of K562 cells into erythrocytes (Figure 7B).

To further verify the effect of DSFF on the differentiation of K562 cells, we detected the effect of CD235a expression in K562 cells, as the elevated expression of CD235a (hemoglobin glycoprotein A) indicates erythrocyte maturation [30]. The results showed that, after DSFF treatment (6.3–50 μg/mL), the expression of CD235a was significantly increased in a concentration-dependent manner, confirming that DSFF induced erythrocyte differentiation (Figure 7C).

CD41 (platelet glycoprotein IIb) is expressed in the megakaryocyte-platelet system, and is an early hematopoietic differentiation antigen [31,32]. Furthermore, the megakaryocyte-platelet differentiation of DSFF-treated K562 cells was detected. As shown in Figure 7D, after DSFF treatment (6.3–50 μg/mL), CD41 expression increased in a concentration-dependent manner, suggesting that DSFF could promote the differentiation of K562 cells into the megakaryocyte-platelet line (Figure 7D). The above results show that DSFF could promote the differentiation of K562 cells into erythrocytes and megakaryocytes-platelet cells for restoring blood cells.

The differentiation of bone marrow cells into mature erythrocytes, megakaryocytes, and platelet cells requires the participation of differentiation and mature-related proteins. The RSK1 p90, c-Myc, and GATA1 proteins play important roles in the proliferation and differentiation of erythrocytes, megakaryocytes, and mast cells [33]. Next, we examined the effect of DSFF on the expression of differentiation and mature-related proteins in K562 cells. The results showed that the expression of RSK1 p90 decreased gradually with the increase in the DSFF concentration (6.3–100 μg/mL), which might promote the proliferation and differentiation of bone marrow cells. Meanwhile, the expressions of c-Myc protein and GATA1 protein were obviously enhanced in a concentration-dependent manner (Figure 7E). The above results show that DSFF might have proliferative promotion and differentiation induction effects on the bone marrow cells, alleviating the myelosuppression and hemocyte abnormity induced by cyclophosphamide.

## 3. Discussion

This study aims to investigate the protective effect of DSFF on cyclophosphamide-induced hematopoietic damage in mice and elucidate the underlying mechanisms. The polysaccharides extracted from brown seaweed were reported to be absorbed into rat [34] and human [35] serum without any change in molecular weight due to intestinal macrophages, liver Kupfer cells [34], or clathrins [36]; thus, stimulating the hematopoietic effect of DSFF (i.p.) in mice was investigated first. Weight gain inhibition, abnormal splenic and thymic indexes, and bone marrow DNA suppression are considered to be serious side effects of high doses of cyclophosphamide. These results suggest that DSFF might have the potential to reverse the weight gain inhibition induced by cyclophosphamide. Interestingly, a single high dose of cyclophosphamide might have different effects on the spleen and thymus, and DSFF might have a strong stimulating effect on the spleen and cause the amelioration of the thymus suppression, which might be associated with the spleen compensatory hyperplasia [37] or splenic immune hyperactivity induced by cyclophosphamide [38]. In addition, the protective effect of DSFF on cyclophosphamide-induced bone marrow DNA inhibition was also observed.

Hematopoietic damage, especially a decrease in the white blood cell number (mainly neutrophils) and platelet abnormalities, are considered to be the main reasons for the poor prognosis of cyclophosphamide. Although fucoidans have the potential to treat chemotherapy-induced leukopenia, fucoidans from different species sources also cause differences in activity. In our present study, we first investigated the stimulating hematopoietic effect of DSFF on white blood cell count (mainly neutrophils) and platelet count. The restorative effects of DSFF on white blood cell count, platelet count, and bone marrow DNA content strongly proved that the DSFF improved the blood routine in mice with cyclophosphamide-induced hematopoietic damage. Consistent with our findings, fucoidans from *Durvillaea antarctica* [39] and *Fucus vesiculosus* [40] were also found to improve hematopoietic damage induced by radiotherapy or chemotherapy at higher concentrations. Both our present work and research carried out by other groups indicate the potential use of fucoidans in improving the blood routine in patients with chemotherapy-induced hematopoietic damage.

Due to the diverse structure of fucoidans, it is difficult to investigate the action mechanisms of DSFF in vivo. As one important subfield of metabolomics, lipidomics studies the overall changes in endogenous lipids and their bioactivity [41]. Endogenous lipids facilitate changes in cellular morphology during division and participate in key signaling events [42]. With the development of high-throughput and high-resolution detection technology and analysis instruments, it is possible to investigate the action mechanisms of polysaccharides using metabolomics methods [43,44]. However, there have been few reports investigating the pharmacological mechanism of polysaccharides based on the lipidomics method. UPLC/Q-Extraction Orbitrap/MS/MS-based lipidomics results revealed 16 potential lipid biomarkers in serum that responded to hematopoietic damage in mice related to the pathways of glycerophospholipid metabolism, linoleic acid metabolism, alpha-linolenic acid metabolism, and arachidonic acid metabolism.

Glycerophospholipid is not only a component of the cell membrane but also plays an important role in regulating immune cell function [45,46]. Some evidence has shown that glycerophospholipid metabolism disorders might cause the abnormal function of bone marrow [47] and affect the percentage of different blood cell types [48].

In the pathways of fatty acid metabolism (*α*-linolenic acid, linoleic acid, and arachidonic acid metabolism), α-linolenic acid (ALA) is the substrate of long-chain, unsaturated omega-3 fatty acids, eicosapentaenoic acid (EPA), eicosapentaenoic acid, and docosahexaenoic acid (DHA) [49]. ALA can promote the phagocytosis of mouse mononuclear macrophages [50], peritoneal macrophages, [51] and promote cell proliferation [52,53]. Furthermore, the DHA synthesized by *α*-linolenic acid is transported into erythrocytes through glucose transporter-1 (GLUT-1) on the erythrocyte membrane, then rapidly reduced in erythrocytes, which helps erythrocytes to maintain a good antioxidant activity [54].

It has been reported that linoleic acid is significant in the regulation of neutrophils [55,56]. Linoleic acid can promote the secretion of immune factors such as IL-6, which can activate the JAK-STAT3 pathway [57], increase the secretion of granulocyte colony-stimulating factor (G-CSF), and ultimately promote the recovery of leukocytes [58].

Arachidonic acid is transformed by phosphatidylcholine under the catalysis of phospholipase A2, which is pivotal in the regulation of immune cells, especially leukocytes and platelets [30,59]. Arachidonic acid is a direct precursor to the synthesis of twenty carbon derivatives; these eicosanoids have important regulatory effects on lipid protein metabolism, white blood cell function, and platelet activation [54].

DSFF exhibited hematopoietic protection in animal experiments, which was supported by data from serum lipidomics. The PCA and OPLS-DA of metabolites in the control group, the model group, and the 3.6 mg/kg DSFF group showed the same results, as the points of each group gathered well and were distributed in different quadrants. Among them, PC (20:1/14:0) and SM (18:0/22:0) are the key lipid molecules through which DSFF exerts protective actions. Consistent with our findings, previous investigations have also shown that Danggui Buxue Decoction could regulate the levels of PC and SM in mice with glycerophospholipid metabolism in hematopoietic damage [60], confirming that lipids (PC and SM) are involved in the mechanism of DSFF. Glycerophospholipid metabolism causes the abnormal function of bone marrow and affects the percentage of different blood cell type; furthermore, glycerophospholipids can also affect cell proliferation, differentiation, and apoptosis [61]. Additionally, our results show DSFF might have proliferative promotion effects and differentiation induction on bone marrow cells, alleviating the myelosuppression and hemocyte abnormity induced by cyclophosphamide. Consistent with our findings, some studies have shown that fucoidan from *Fucus vesiculosus* [62] and *Chordaria flagelliformis* [63] could also promote hematopoietic function in hematopoietic damaged animals by promoting the proliferation and differentiation of hematopoietic cells. Moreover, our present work also proved that the lipidomics analysis method might provide a good methodological reference for elucidating the complex mechanism of polysaccharides through studying the overall changes of endogenous lipids [41]. While oral administration is the usual route of drug administration, fucoidans have been reported to be degraded by the gut microbiota [64,65], which might affect their stimulating hematopoietic effect. Anisimova et al. found that fucoidan administered by subcutaneous injection would stimulate the hematopoiesis of mice with cyclophosphamide-induced hematopoietic damage [11]; in addition, Lee et al. indicated that fucoidan administered by intraperitoneal injection also stimulated the hematopoiesis of mice with hematopoietic damage [63]. Therefore, we speculate that intraperitoneal or subcutaneous injection might be the optimal administration mode of fucoidan for hematopoiesis stimulation. Of course, the intestinal microbiota might be a target of fucoidans, in addition to its direct role in cells [66]. Although we initially tried to explore and verify the mechanisms of DSFF on related pathways through the above experiments, the question of at which molecular weight fucoidan plays a role and other more detailed mechanisms of DSFF need to be further investigated.

## 4. Materials and Methods

### 4.1. Materials and Reagents

Simulated digestive product of fucoidan from *Sargassum fusiforme* (DSFF) was prepared according to our laboratory method [12]. In brief, the main composition of DSFF was total sugar (51.01%) with ignorable protein and uronic acid and the sulfate content was 16.99%. Lipopolysaccharide was not detected in DSFF using the test-tube quantitative chromogenic matrix method according to the instructions of the Limulus Endotoxin Detection Kit (Xiamen Bioendo Technology Co., Ltd., Xiamen, China). The analysis of the monosaccharide composition suggested that DSFF was fucoidan with abundant fucose (36.05%), galactose (18.60%), glucose (7.99%), xylose (5.64%), mannose (4.57%), mannuronic acid (4.26%), glucosamine (2.68%), and glucuronic acid (1.62%). The molecular weight distribution (area percentage %) of DSFF was 146.1 (21.0%), 88.3 (16.5%), 42.4 (16.1%), 28.5 (10.2%), and 25.9 (23.2%) kilodaltons.

Chronic myeloid leukemia cells K562 were purchased from the Cell Bank of the Chinese Academy of Sciences (Shanghai, China). Roswell Park Memorial Institute (RPMI) 1640 medium was purchased from Genio Biopharmaceutical Technology Co., Ltd. (Hangzhou, China). Limulus endotoxin detection kit was purchased from Xiamen Bioendo Technology Co., Ltd. (Xiamen, China). Fetal bovine serum (FBS) was purchased from Shanghai Jitai Biotechnology Co., Ltd. (Shanghai, China). Cyclophosphamide for injection was purchased from Jiangsu Shengdi Pharmaceutical Co., Ltd. (Lianyungang, China). Bovine serum albumin (BSA) was purchased from Beijing Solai Bao Technology Co., Ltd. (Beijing, China). Recombinant human granulocyte colony stimulating factor injection (rhG-CSF) was purchased from Qilu Pharmaceutical Co., Ltd. (Jinan, China). Benzidine, acetic acid, 30% H_2_O_2_, and sodium nitroferricyanide dihydrate were purchased from Sinopharm Chemical Reagents Co., Ltd. (Beijing, China). FITC-anti-CD235A antibody and APC-anti-CD41 were purchased from BioLegend (San Diego, CA, USA). RSK1 p90, c-Myc, GATA1, and GAPDH were purchased from Shenyang Wanlei Biological Technology Co., Ltd. (Shenyang, China). PC (14:0/18:1), LysoPC (14:0/0:0), LysoPC (16:0/0:0), LysoPC (18:1/0:0), PG (16:0/18:1), LysoPG (16:0/0:0), LysoPG (18:1/0:0), LysoPS (18:0/0:0), PS (16:0/18:1), LysoPE (16:0/0:0), PA (16:0/18:1), and PI (16:0/18:1) were purchased from AVANTI polar lipid (Alabaster).

### 4.2. Hematopoietic Damage Model Establishment and Treatment

SPF-grade female Kunming mice (18–20 g, 4–6 weeks old) were purchased from Jinan Pengyue Experimental Animal Breeding Co., Ltd., license number: SCXK (Lu) 20190003. Then, they were randomly divided into a control group (*n* = 10) and experimental group (*n* = 50) according to body weight after adaptive feeding for 7 days. Hematopoietic damage model mice were established by the tail intravenous injection of cyclophosphamide (250 mg/kg) once in the experimental group mice. Then, the successive experimental mice were divided into a model group (*n* = 10), rhG-CSF group (*n* = 10), low-dose-DSFF group (*n* = 10), medium-dose-DSFF group (*n* = 10), and high-dose-DSFF group (*n* = 10), according to their significantly reduced white blood cell count (WBC) and were treated with normal saline (i.p.), rhG-CSF (s.c., 22.5 μg/kg), low-dose DSFF (i.p., 1.8 mg/kg), medium-dose DSFF (i.p., 3.6 mg/kg), and high-dose DSFF (i.p., 7.2 mg/kg), respectively. The control group was treated with the same volume of normal saline (0.1 mL/10 g body weight). Animal experiments were approved by the Institutional Animal Care and Use Committee of the Ocean University of China (No.OUC-SMP-2020-05-01).

### 4.3. Detection of Relevant Indicators

#### 4.3.1. Examination of Blood Routine

From the first day after administration, 20 μL of blood was collected from tail vein of each mouse on the 1st, 3rd, 5th, and 7th day. The values of white blood cells, neutrophils, and platelets in each group were detected on the Celltac E blood cell analyzer (Nihon Kohden, Japan), and the body weight of the mice was recorded. The weight of spleen and thymus of mice was determined at the end of the experiment.

#### 4.3.2. Detection of DNA Content from Bone Marrow

DNA content was detected as reported previously [67]. In brief, after euthanasia, the right femur of mice in each group was taken. All the bone marrow was flushed with 10 mL 0.005 M CaCl_2_ into a clean centrifuge tube, and placed in a refrigerator at 4 °C for 30 min, and then centrifuged at 2750× *g* for 15 min. The supernatant was discarded and replaced with 5 mL of 0.2 nM HClO_4_ solution. DNA content was detected by ultra-micro spectrophotometer (SpectraMax M5 and M5e Multi-Detection Microplate Readers, 206 Molecular Devices, Sunnyvale, CA, USA.) after the cooling and filtration of the solution.

### 4.4. Serum Samples Preparation for Lipidomics

After the last administration on the 8th day, blood was collected from the eyeball within 30 min, placed at 4 °C for 30 min, and centrifuged at 3300× *g* for 15 min. The serum of each group was carefully aspirated and stored at −80 °C for further use.

Serum samples from each group were gradually thawed and centrifuged at 4 °C 4400× *g* for 10 min. Then, 50 μL serum was added to 200 μL methanol containing internal standards (2-chloro-L-phenylalanine and L-theanine), vortexed for 2 min, then centrifuged at 4 °C 15,400× *g* for 10 min to precipitate protein. After the drying of the supernatant, 200 μL precooling methanol-water (1:1, *v/v*) solution was added, vortexed for 2 min, and centrifuged at 4 °C at 15,950× *g* for 10 min. Then, we transferred the supernatant into the sample vial for LC-MS analysis. The QC sample was prepared by taking 10 μL of serum from each group, mixing it, and then following the above steps for the serum samples.

### 4.5. UPLC Q-Exactive Orbitrap MS/MS Analysis

All samples were investigated in a Thermo UltiMate 3000. The following were the chromatographic conditions: Agilent InfinityLab Poroshell 120 (2.1 × 5 mm, 1.7 μm), Agilent InfinityLab Poroshell 120 EC-C18 (2.1 × 100 mm, 1.7 μm); the mobile phase was 0.1% formic acid aqueous solution (A)-0.1% formic acid acetonitrile solution (B) containing 10 mM ammonium acetate. The column temperature was as follows: 35 °C; flow rate: 0.3 mL/min; sample size: 5 μL. The elution procedure was as follows: 0–2 min, 98–90% A; 2–6.5 min, 90–58% A; 6.5–12 min, 58–10% A; 12–14 min, 10–5% A; 14–15.5 min, 5% A; 15.5–16 min, 5–98% A; 16–20 min, 98% A.

The following were the mass spectrometric conditions: ESI ion source, positive and negative ions full MS/dd-MS^2^ mode detection; sheath gas volume flow: 35, auxiliary gas volume flow: 10 psi; spray voltage: 3.0 kV; capillary temperature: 350 °C, auxiliary temperature: 300 °C. Full MS mass scanning range *m/z*: 80–1200, detection resolution: 70,000 FWHM, dd-MS^2^ normalized collision energy NCE set 30, 40, and 50. Higher Energy Collision Induced Dissociation (HCD) was used for the secondary mass spectrometry with a resolution of 17,500.

The processed serum data matrix and UPLC-MS/MS raw data files were imported into Progenesis QI (Wasters, MA, USA) for matching, alignment, and normalization; then, the processed data were imported into SIMCA-P 14.1 (Umetrics, Sweden) for multivariate statistical analysis, including principal component analysis (PCA), to preliminarily observe the separation of samples and eliminate abnormal samples. Orthogonal partial least-squares discrimination analysis (OPLS-DA) was performed for the comparison of mass spectrometry between the control group and model group. Potential endogenic metabolites were screened according to the standard of VIP > 1, *p* < 0.05 and compared with databases (KEGG, HMDB, Lipid Maps, and Massbank) to determine the differential metabolite information. Finally, the selected metabolites were imported into MetaboAnalyst (https://www.metaboanalyst.ca/faces/home.xhtml, accessed on 12 February 2021) to perform the metabolic pathway analysis.

### 4.6. K562 Cell Proliferation Assay

K562 cells (5000 cells/well) were treated with different concentrations (6.3–100 μg/mL) of DSFF for 48 h; then, we added 10 μL CCK-8 solution to incubate for 4 h. The absorbance of each well was measured by an ultra-micro spectrophotometer at 450 nm (SpectraMax M5 & M5e Multi-Detection Microplate Readers, 206 Molecular Devices, Sunnyvale, CA, USA).

### 4.7. Differentiation Assay of K562 Cells Using Benzidine Staining and Flow Cytometry

K562 cells (2 × 10^5^ cells/well) were treated with different concentrations (6.3–50 μg/mL) of DSFF for 48 h. Then, the cells were collected; washed with cold PBS twice; added with 14 μL 0.4% benzidine solution, 1 μL 12% acetic acid solution, and 1 μL 30% H_2_O_2_ solution. This was followed by incubating them for 5 min and then adding 1 μL sodium nitroso ferricyanide solution. After incubation for 30 min, the number of blue-positive cells in 300 cells was recorded by an XD-inverted microscope (Sunny Optical Technology (Group) Co., Ltd., Shanghai, China).

K562 cells (3 × 10^5^ cells/well) were treated with different concentrations (6.3–50 μg/mL) of DSFF for 48 h, collected, washed with cold PBS twice, and added to 0.9 μL of 0.5 mg/mL FITC-anti CD235a antibody or APC-CD41 antibody. After incubation for 15 min, the CD235a and APC-CD41 expression levels in K562 cells were detected by a flow cytometer (Beckman Coulter MoFlo XDP, Fullerton, CA, USA).

### 4.8. Western Blotting Detects Differentiation-Related Protein Expression in K562 Cells

K562 cells (5 × 10^5^ cells/well) were treated with different concentrations (6.3–100 μg/mL) of DSFF for 48 h. Proteins were collected, quantified, separated through electrophoresis, and transferred to a PVDF membrane. The membrane was blocked for 2 h, then incubated overnight with primary antibody solution containing different proteins (RSK1 p90, c-Myc, and GATA1, 1:500) at 4 °C, blocked for 2 h, and incubated with secondary antibody solution for 2 h. *β*-actin was used as an internal reference to detect the expression of specific proteins. Specific proteins were detected with enhanced chemiluminescence by FluorChem E (Protein Simple, San Jose, CA, USA).

### 4.9. Statistical Analysis

Quantitative experimental data were expressed as mean ± SD. Statistical tests were carried out with the GraphPad Prism 8 software (GraphPad Software, San Diego, CA, USA). Statistical analyses between multiple groups were performed using GraphPad Prism 8 by one-way analysis of variance plus the multiple-comparisons Tukey’s test (GraphPad Software, San Diego, CA, USA). *p* < 0.05 was considered statistically significant.

## 5. Conclusions

In summary, our present study, for the first time, revealed that DSFF exerted a hematopoietic damage protection effect on mice with cyclophosphamide-induced hematopoietic damage via restoring white blood cells, neutrophils, platelets, as well as bone marrow cells. Moreover, we first illustrated that the mechanisms underlying the hematopoietic damage protection of DSFF might mainly act through improving the imbalanced lipid metabolism and ultimately promoting bone marrow cell proliferation and differentiation. The current findings indicated that DSFF showed good potential and application value in alleviating hematopoietic damage and provided a greater understanding of the pharmacological function and mechanisms of fucoidans.

## Figures and Tables

**Figure 1 marinedrugs-20-00201-f001:**
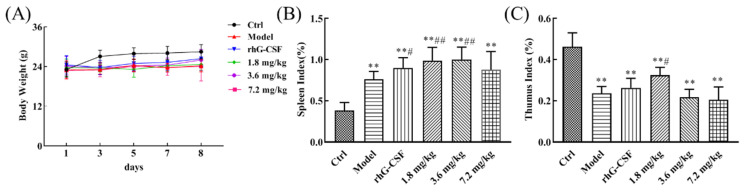
Effects of simulated digestive product of fucoidan from *Sargassum fusiforme* (DSFF) on the (**A**) body weight gain, (**B**) splenic index, and (**C**) thymic index of mice with cyclophosphamide-induced hematopoietic damage. ** *p* < 0.01, vs. control; ^#^
*p* < 0.05, ^##^ *p* < 0.01, vs. model.

**Figure 2 marinedrugs-20-00201-f002:**
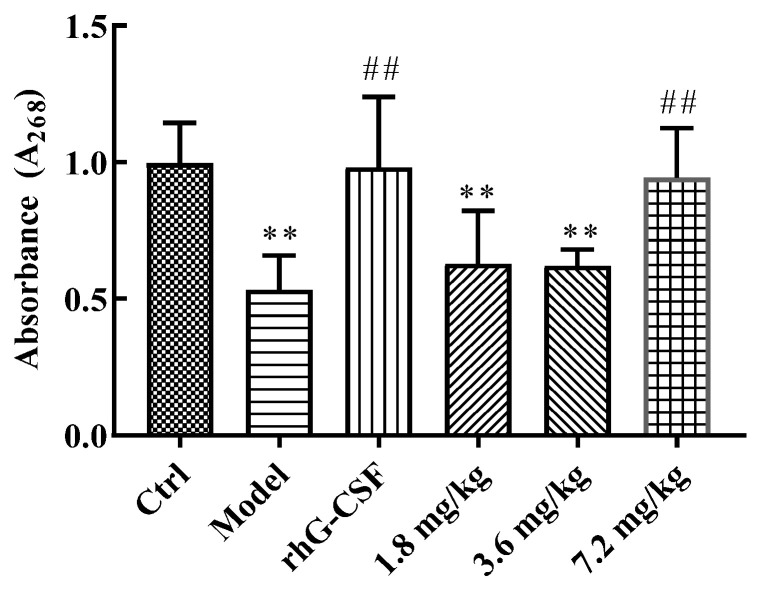
Effects of DSFF on bone marrow DNA content in cyclophosphamide-induced mice. Data are the mean ± standard deviation (*n* = 10). ** *p* < 0.01, vs. control; ^##^ *p* < 0.01, vs. model.

**Figure 3 marinedrugs-20-00201-f003:**
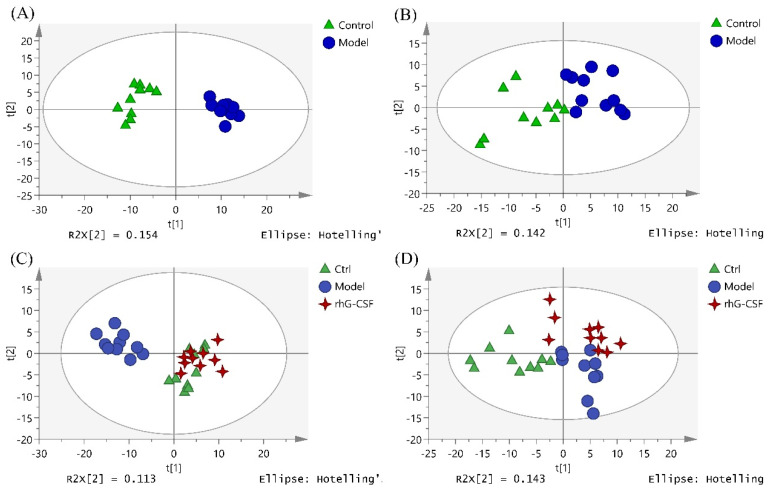
PCA score plots of serum samples: (**A**) control vs. model in positive ion mode; (**B**) control vs. model in negative ion mode; (**C**) control vs. model vs. rhG-CSF in positive ion mode; (**D**) control vs. model vs. rhG-CSF in negative ion mode.

**Figure 4 marinedrugs-20-00201-f004:**
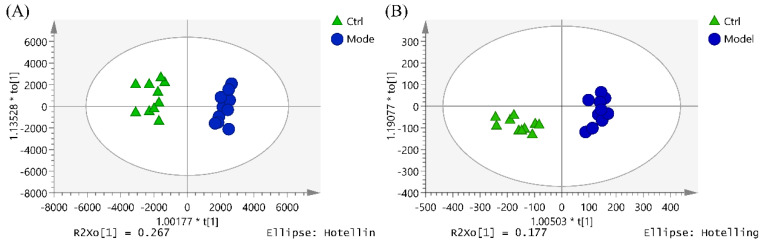
OPLS-DA score plot of serum samples from control group and model group in (**A**) positive ion mode and (**B**) negative ion mode.

**Figure 5 marinedrugs-20-00201-f005:**
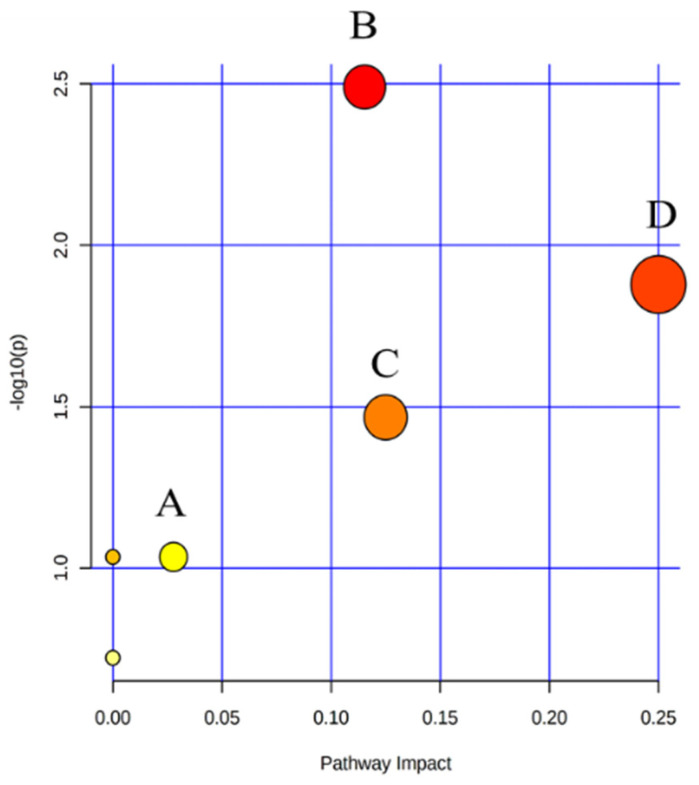
Differential metabolite pathway analysis of serum samples in the control group and the model group. A: arachidonic acid metabolism; B: glycerophospholipid metabolism; C: alpha-linolenic acid metabolism; D: linoleic acid metabolism.

**Figure 6 marinedrugs-20-00201-f006:**
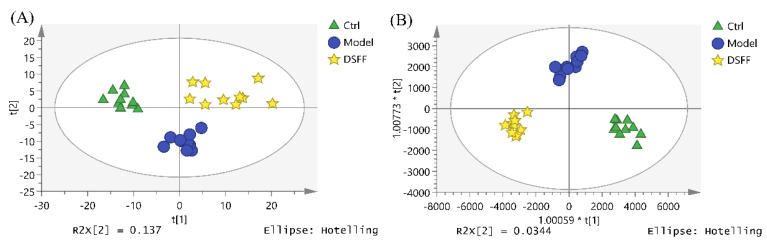
PCA and OPLS-DA score plots of serum samples between the control group, model group, and 3.6 μg/mL DSFF group under positive ion mode. (**A**): PCA score plot; (**B**): OPLS-DA score plot.

**Figure 7 marinedrugs-20-00201-f007:**
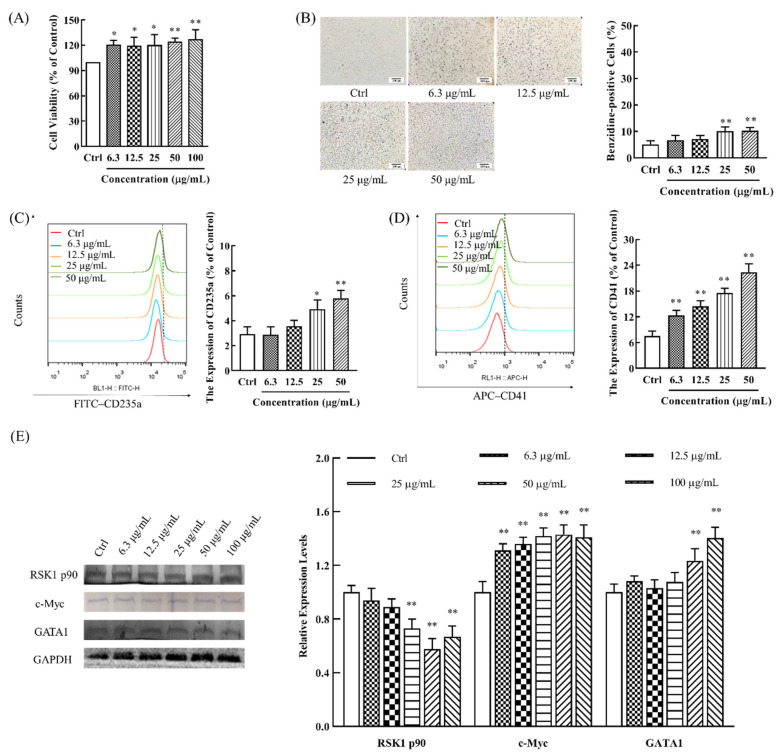
Effects of simulated digestive product of fucoidan from *Sargassum fusiforme* (DSFF) on the proliferation and differentiation of K562 cells. (**A**) Proliferative effect of DSFF on K562 cells using CCK-8 assay; (**B**) effects of DSFF on the differentiation of K562 cells using a benzidine-staining assay (10×, scale bar: 100 μm); (**C**) effects of DSFF on the erythroid differentiation of K562 cells using flow cytometry; (**D**) effects of DSFF on the megakaryocyte differentiation of K562 cells using flow cytometry; (**E**) effects of DSFF on the differentiation-related protein expression of K562 cells using Western blotting. Data are the mean ± standard deviation (*n* = 10). * *p* < 0.05, ** *p* < 0.01, vs. control.

**Table 1 marinedrugs-20-00201-t001:** Effects of simulated digestive product of fucoidan from *Sargassum fusiforme* (DSFF) on white blood cell, neutrophil, and platelet counts in mice with cyclophosphamide-induced hematopoietic damage.

Groups	First Day	Third Day	Fifth Day	Seventh Day
	White Blood Cell Count (10^9^/L, Mean ± SD)			
Control	11.29 ± 2.83	12.21 ± 2.92	10.69 ± 3.81	14.55 ± 3.38
Model	4.46 ± 1.41 **	2.04 ± 1.16 **	3.85 ± 1.85 **	16.34 ± 3.58
rhG-CSF	4.48 ± 1.40 **	1.61 ± 0.52 **	11.03 ± 3.88 ^##^	26.81 ± 4.31 **^##^
1.8 mg/kg	4.50 ± 1.41 **	1.91 ± 0.72 **	10.21 ± 4.45 ^##^	22.44 ± 3.82 **^##^
3.6 mg/kg	4.50 ± 1.37 **	2.08 ± 0.78 **	8.39 ± 2.42 ^##^	18.20 ± 2.95
7.2 mg/kg	4.54 ± 1.46 **	1.72 ± 0.65 **	8.36 ± 3.40 ^##^	15.99 ± 2.57
Groups	Neutrophil Count (10^9^/L, Mean ± SD)			
Control	4.21 ± 1.53	4.83 ± 1.57	3.76 ± 1.14	4.41 ± 1.19
Model	2.22 ± 0.98 **	0.47 ± 0.38 **	2.31 ± 1.38 **	15.25 ± 4.01 **
rhG-CSF	2.09 ± 0.83 **	0.57 ± 0.42 **	9.25 ± 3.76 **^##^	24.11 ± 6.40 **^#^
1.8 mg/kg	2.03 ± 1.29 **	0.42 ± 0.27 **	7.86 ± 3.96 **^##^	20.28 ± 8.93 **^#^
3.6 mg/kg	2.26 ± 0.97 **	0.35 ± 0.18 **	6.39 ± 2.38 ^##^	16.65 ± 5.27 **
7.2 mg/kg	2.43 ± 1.62 **	0.63 ± 0.66 **	6.43 ± 3.05 ^##^	15.03 ± 4.33 **
Groups	Platelet Count (10^9^/L, Mean ± SD)			
Control	770.67 ± 58.75	758.07 ± 71.77	755.40 ± 135.06	758.20 ± 66.65
Model	769.33 ± 91.33	498.47 ± 70.46 **	359.27 ± 99.7 **	638.86 ± 179.33
rhG-CSF	813.93 ± 115.15	499.6 ± 91.21 **	360.33 ± 88.62 **	709.87 ± 117.90
1.8 mg/kg	794.53 ± 58.65	445.93 ± 112.51 **	422.73 ± 69.20 **	850.71 ± 144.16 ^#^
3.6 mg/kg	815.33 ± 116.58	467.8 ± 99.03 **	362.00 ± 106.44 **	771.80 ± 180.70
7.2 mg/kg	858.40 ± 118.22	404.73 ± 124.02 **	347.93 ± 116.03 **	849.2 ± 256.08 ^#^

Data are the mean ± standard deviation (*n* = 10). ** *p* < 0.01, vs. control; ^#^ *p* < 0.05, ^##^ *p* < 0.01, vs. model.

**Table 2 marinedrugs-20-00201-t002:** Results of identified key potential biomarkers (M: model group; C: control group; R: rhG-CSF group; D: DSFF group).

No.	t_R_/min	*m/z*	HMDB ID	KEGG	Formula	Adduct	Identification	M vs. C	R vs. M	D vs. M
1	11.98413	496.3399	HMDB0061709	-	C_24_H_50_NO_7_P	M+H, M+Na	2-palmitoyl-sn-glycero-3-phosphocholine	↓		↓
2	12.46675	508.3765	HMDB0013122	C04230	C_26_H_54_NO_6_P	M+H, M+Na	LysoPC (P-18:0)	↓	↑	
3	13.00928	806.5689	HMDB0007991	C00157	C_46_H_80_NO_8_P	M+H	PC (16:0/22:6)	↓		
4	13.68895	327.2332	HMDB0002183	C06429	C_22_H_32_O_2_	M-H	Docosahexaenoic acid	↓		
5	14.0069	810.6000	HMDB0008464	C00157	C_46_H_84_NO_8_P	M+H	PC (20:4/18:0)	↓		↓
6	14.58425	465.3047	HMDB0000653	C18043	C_27_H_46_O_4_S	M-H	Cholesterol sulfate	↓		
7	14.57367	760.5846	HMDB0008295	C00157	C_42_H_82_NO_8_P	M+H	PC (20:1/14:0)	↓		↑
8	16.48282	768.5891	HMDB0013407	-	C_44_H_82_NO_7_P	M+H	PC (16:0/20:4)	↓		
9	16.5502	812.6139	HMDB0008399	C00157	C_46_H_86_NO_8_P	M+H-H_2_O, M+H	PC (20:3/18:0)	↓	↑	
10	16.7293	742.5735	HMDB0008159	C00157	C_42_H_80_NO_7_P	M+H	PC (18:2/P-16:0)	↓		
11	16.8413	811.6675	HMDB0012091	C00550	C_45_H_93_N_2_O_6_P	M+Na	SM (d18:0/22:0)	↓	↑	↑
12	17.3122	806.569	HMDB0008339	C00157	C_46_H_80_NO_8_P	M+H	PC (20:2/18:4)	↓		
13	17.42562	780.5533	HMDB0008495	C00157	C_44_H_78_NO_8_P	M+H	PC (20:5/16:0)	↓		↓
14	17.44898	756.553	HMDB0008199	C00157	C_42_H_78_NO_8_P	M+H	PC (18:3/16:0)	↓		
15	17.78938	627.5344	HMDB0007171	-	C_41_H_72_O_5_	M+H-H_2_O	DG (18:0/20:4)	↓		↓
16	18.01642	806.5687	HMDB0008212	C00157	C_46_H_80_NO_8_P	M+H	PC (18:3/20:3)	↓		

Note: “↑” and “↓” represent higher and lower content expression between two groups.

## Data Availability

Not applicable.

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
