# Peer review of "Stimulating the Hematopoietic Effect of Simulated Digestive Product of Fucoidan from Sargassum fusiforme on Cyclophosphamide-Induced Hematopoietic Damage in Mice and Its Protective Mechanisms Based on Serum Lipidomics"

_marinedrugs, 2022, doi:10.3390/md20030201_

Round 1

Reviewer 1 Report

  • Title very confusing
  • If DSFF is ab acronym for “biomimetic hydrolysis fucoidan from Sargassum fusiforme”, shouldn’t it be BSFF?
  • Figure 1A is difficult to read, with the error bars superimposing each other
  • The text explaining Figure 1A (starting with “Cytotoxic drug-induced weight loss is…” should elaborate a bit more to explain what is being observed. I had to go back and forth between the text and the figure several times to understand what was happening.
  • Also, this text says: “As shown in Fig. 1A, cyclophosphamide induced the significant decrease of the body weight in model group”. However, figure 1A show the weight rising slightly above 23 grams after 8 days. The control group, which I imagine received no cyclophosphamide, increased weight from 23 to ~29g, so the figure expresses not a loss of weight, but a lack of gain in weight.
  • The paragraph explaining splenic and thymic indexes (figures 1B and 1C) is also very confusing, should be re-written in more detail, and expressing the meaning of that data in relationship with the control.
  • The manuscript implies that the adjuvants DSFF and rhG-CSF were ministered concomitantly with cyclophosphamide, but this is not clearly indicated in the text.
  • Much of the scientific information is hard to follow due to the prolixity of some parts of the text, further complicated by poor grammatical construction and lack of details in some of the text explaining figures.
  • I would strongly suggest this manuscript to be edited/re-written by a fluent English writer/speaker.

Author Response

Dear editors,

We want to thank the editors and reviewers for spending their valuable time reviewing and providing constructive suggestions. The detailed responses to the reviewers are in the attachment.

Reviewer 2 Report

This paper describes the use of a partially depolymerised fucoidan, using a technique that mimics human digestion, to inhibit chemotherapy effects on hematopoeisis in a mouse model.  It also examines serum lipidomics as a marker system, which is laudable and interesting.

Overall, I believe the paper is worthy of publication, and adds to the body of the literature in a useful way.   The protective effects of i.p dosed fucoidan are worthy of investigation.

Firstly, I wanted to address the use of the word ‘biomimetic’.  I understand that by this, the authors refer to the ‘digestion similar’ technique they use to make the partially depolymerised fucoidan.   In the text, the word ‘biomimetic’ becomes confusing, as the reader can assume they mean that the fucoidan is a biomimetic of some other compound.    I wonder whether it may be better to use another term- just ‘partially hydrolysed’ would be fine.

In the abstract, I suggest quoting the dose ranges assessed, and clarifying that the dosing is I.P .

In the results, I particularly noted that your lowest dose had already achieved the maximum effect in terms of WBC count, platelet count etc.  I think this needs a mention your results- the lowest dose is sufficient for restoration of some of the functions, although the highest dose was need to restore bone marrow DNA measures.

In the discussion, I think it is particularly relevant to mention oral dosing, in comparison to ip or Sc dosing. Route of  administration is important to future application, oral usually being the preferred route. 

 For example, Anisimova used subcutaneous dosing in their cyclophosphamide model, whereas Lee used IP.    I do not think that oral dosing has been assessed in this model to date.

A minor English check throughout would be appropriate.   

Some key sentences need addressing:

Line 48 Fucoidan is a unique acidic polysaccharide containing sulfuric acid group, which con- 48

tributes to its applaudable biological functions

please replace with something simple eg

Fucoidans are a class of fucose rich sulphated polysaccharides.

Line 51 misspelled haematopoiesis

Line 204 ‘mouse serum’’

Line 232 should read ‘After treatment with different concentrations of ….

Author Response

(The authors gave the same response as above.)

Reviewer 3 Report

The authors found that DSFF, like several other fucoidans, resulted in a rapid recovery of hematopoietic damage in mice induced
 cyclophosphamide. The authors concluded that
 mechanisms of protection of hematopoietic damage in DSFF are associated with improvement of unbalanced elevated lipids and stimulation of proliferation and differentiation of bone marrow cells. These results indicate the prospects for further study of DSFF as a drug for the treatment and prevention of hematopoietic damage during high-dose and high-intensity chemotherapy in cancer patients with neutropenia and pancytopenia.

чHowever, when analyzing the results, a number of questions arise.
Why does the spleen index increase and the thymus index decrease during the administration of fucoidan?
How to explain the lack of effect of fucoidan at a concentration of 7.2 µg/ml, does this indicate the absence of a dose-dependent effect?

The conclusions of the authors about the stimulation of differentiation of precursors of erythrocytes and megakaryocytes cause doubts.

Is it possible to consider an increase in the level of СD235a + cells from 3% to 6% as evidence of differentiation of K 562 cells and how does this correlate with bone marrow progenitor cells?

How, under the influence of fucoidan, increased proliferation and differentiation of K 562 cells are combined?

Seaweed is known to contain gram-negative commensal bacteria and therefore seaweed extracts may be contaminated with LPS. LPS, in turn, is able to cause, just like fucoidans, an emergency stimulation of hematopoiesis. Therefore, it is important to indicate the content or absence of FPS in the DSFF.

Author Response

(The authors gave the same response as above.)

Round 2

Reviewer 1 Report

Most of the suggested modifications were done. One of the figures is still difficult to see, but the overall trend is discernible.

Author Response

Dear editors and reviewers,

We want to thank the editors and reviewers for spending their valuable time reviewing and providing constructive suggestions. The detailed responses to the reviewers are in the attachment.

Reviewer 3 Report

Dear Editor,
The authors of the revised manuscript and provided answers to the questions posed. The presented version of the article can be recommended for publication. However, the question of possible contamination of fucoidan with LPS remained unanswered. This question is important, since fucoidans, like LPS, have the effect of emergency stimulation of hematopoiesis. Therefore, the authors should indicate the possible content of LPS or its absence due to chemical purification in fucoidan.

Author Response

(The authors gave the same response as above.)
